# Predictive factors of venous recanalization in upper-extremity vein thrombosis

**Gaëtan Ploton[1], Nicolas Brebion[2], Béatrice Guyomarch[3], Marc-Antoine Pistorius[1,4], Jérôme Connault[1,4], Jeanne Hersant[1], Alizée Raimbeau[1,4], Guillaume Bergère[1,4], Mathieu Artifoni[1,4], Cécile Durant[1,4], Giovanni Gautier[1,4], Romain Dumont[4,5], Jean-Manuel Kubina[2], Claire Toquet[6,7☯, Olivier Espitia[1,4,7]☯ ***

**1** Department of Internal Medicine and Vascular Medicine, CHU de Nantes, Nantes, France, **2** Department of Vascular Medicine, CHD La Roche-sur-Yon, La Roche-sur-Yon, France, **3** Direction de la recherche, Plateforme de Méthodologie et Biostatistique, CHU de Nantes, Nantes, France, **4** UNAV, Nantes Vascular Access Unit, CHU de Nantes, Nantes, France, **5** Department of Anesthesia and Critical Care, CHU de Nantes, Nantes, France, **6** Department of Pathology, CHU de Nantes, Nantes, France, **7** Université de Nantes, CHU de Nantes, Nantes, France

☯ These authors contributed equally to this work.
\* olivier.espitia@chu-nantes.fr

## Abstract

### Background

Upper extremity venous thrombosis (UEVT) represents about 10% of venous thrombo-embolic disease. This is mainly explained by the increasing use of central venous line, for oncologic or nutritional care. The factors associated with venous recanalization are not known.

### Objective

The aim of this study was to investigate prognosis factor associated with venous recanalization after UEVT.

### Methods

This study included patients with UEVT diagnosed with duplex ultra-sonography (DUS) from January 2015 to December 2017 with DUS evaluations during follow-up. A multivariate Cox proportional-hazards-model analysis was performed to identify predictive factors of UEVT complete recanalization.

### Results

This study included 494 UEVT, 304 proximal UEVT and 190 distal UEVT. The median age was 58 years, 39.5% were women. Clinical context was: hematological malignancy (40.7%), solid cancer (14.2%), infectious or inflammatory context (49.9%) and presence of venous catheters or pacemaker leads in 86.4%. The rate of recanalization without sequelae of UEVT was 38%. For all UEVT, in multivariate analysis, factors associated with complete vein recanalization were: thrombosis associated with central venous catheter (CVC) (HR:2.40, [1.45;3.95], p<0.001), UEVT limited to a venous segment (HR:1.94, [1.26;3.00],

**Funding:** OE received mobility grant, from Société Française de Médecine Vasculaire.

**Competing interests:** The authors have declared that no competing interests exist.

p = 0.003), occlusive thrombosis (HR:0.48 [0.34;0.67], p<0.0001), the presence of a PICC Line (HR:2.29, [1.48;3.52], p<0.001), a thrombosis of deep and distal topography (HR:1.70, [1.10;2.63], p = 0.02) or superficial thrombosis of the forearm (HR:2.79, [1.52;5.12], p<0.001). For deep and proximal UEVT, non-occlusive UEVT (HR:2.23, [1.49;3.33], p<0.0001), thrombosis associated with CVC (HR:1.58, [1.01;2.47], p = 0.04) and infectious or inflammatory context (HR:1.63, [1.10;2.41], p = 0.01) were factors associated with complete vein recanalization.

## Conclusion

In this study, factors associated with UEVT recanalization were UEVT limited to a venous segment, thrombosis associated with CVC, a thrombosis of deep and distal thrombosis topography and superficial thrombosis of the forearm. Occlusive thrombosis was associated with the absence of UEVT recanalization.

## Introduction

Upper extremity venous thrombosis (UEVT) is an increasingly venous thromboembolic disease (VTED). Today, UEVT represent 10% of all venous thrombosis, whereas it represented 1 to 4% in the 2000s [1, 2]. This is mainly explained by the increasing use of peripherally inserted central catheter -line (PICC-LINE) and of central venous catheter (CVC).

The major UEVT-related diseases and conditions are: venous catheter [3], a solid neoplasia [4], a hematological malignancy [5], a thoracic outlet syndrome (TOS) [6], an estrogenic hormonal impregnation, hereditary or acquired biological thrombophilia (APLS) [7], acute kidney failure or kidney failure requiring dialysis [8] or other situations such as flares of inflammatory diseases [9].

The epidemiology, pathophysiology, treatment and management of UEVT [10], although much less studied than the lower extremities, have long been considered like lower extremity vein thrombosis (LE-VT). However, it is a particular form of VTED: diagnostic elements, clinical features, risk factors and recurrent VTED risks seem different between these two types of thrombosis [11]. UEVT characteristics are poorly known, only a few studies have studied UEVT and they only concerned deep thromboses [12–17]. The factors associated with venous recanalization of UEVT and thrombotic recurrences are also poorly known. Moreover, in patients with chronic diseases such as cancer, it is important to preserve venous capital of the upper limbs to allow treatments administration.

The objective of this study was to describe the predictive factors of venous recanalization in a large cohort of patients with UEVT [5].

## Methods

This retrospective and monocentric study included patients with UEVT, diagnosed between January 1, 2015 and December 31, 2017, defined on duplex ultrasonography (DUS) by a hypo or isoechoic endoluminal picture, without Doppler flow, and/or with incompressibility of the vein. UEVT linked to the CVC was defined by a blood clot facing the catheter with blood clot in contact with the vein wall with a length > 5 mm [5].

The included patients had to have had at least one DUS control of the UEVT during follow-up.

The location of UEVT is described according to the most proximal thrombosed venous segment. The innominate, internal jugular, subclavian and axillary veins belong to the deep and proximal venous network. The humeral, ulnar and radial veins belong to the deep and distal network. Cephalic and basilic veins belong to the superficial network.

The data were collected using a standardized collection grid. Minor or major bleeding was defined according to International Society of thrombosis and Hemostasis (ISTH) criteria [18, 19].

Pulmonary embolism (PE) at diagnosis was diagnosed by CT angiography or by pulmonary ventilation scintigraphy between UEVT diagnosis until 14 days before UEVT diagnosis.

At each DUS follow-up, thromboses were categorized as extending, stable or regressing and the absence or the type of venous sequelae (wall thickening> 4mm, persistence of occlusive thrombosis) were recorded.

UEVT sequelae were defined as a persistent occlusive thrombosis, presence of vein synechia with vein wall thickening> 4mm or vein shrinkage.

Patients for whom the computerized medical record was incomplete and isolated superior vena cava thrombosis were excluded.

The study was approved by the ethics committee of Nantes University Hospital (GNEDS) and complies with the requirements of the National Commission for Computing and Liberties, in accordance with current French legislation. Each patient included in this study received written information and no patient objected to this study.

Quantitative values were expressed in terms of counts and percentages. The mean or median comparisons were made using t-test or Mann Whitney test. Frequency comparisons by a Chi square test or a Fisher exact test according to the statistical headcount. Prognostic factors associated with complete UEVT recanalization were evaluated with Cox models. Hazard Ratios (HR) with their 95%CI has been estimated as association measures. Variables with p < 0.05 in univariate model and all the variables already known to be confounding factors were candidate variables for the first multivariate model (manual step by step selection model). Survival curves were estimated with their 95% confidence interval (95%CI) using Kaplan-Maier estimators and Log rank tests were performed to compare complete UEVT recanalization free survival between groups. SAS version 9.4 software was used to perform analyzes.

The data underlying the results presented in the study are available on S1 File.

## Results

This study included 494 patients; 304 (61.5%) proximal UEVT and 190 (38.5%) distal UEVT (Fig 1).

Patient's and UEVT characteristics and evolution are presented in Table 1. The most frequent solid neoplasia was colonic or bowel cancer (n = 17, 24.3% of solid cancer) and the most frequent hematological malignancy was acute leukemia (n = 101, 50.2% of hematological malignancy); 53 UEVT occurred in a context of renal failure (10.8%) and 3 in patients with inherited thrombophilia (0.6%).

Regarding the presence of an endovenous device, the most frequent devices were PICC-LINE (n = 177, 41.5% of the devices); peripheral venous catheter (PVC) (n = 66, 15.5%); CVC (n = 64, 15.0%); implantable ports (n = 57, 13.4%); dialysis catheters (n = 31, 7.3%); Pace-Maker (PM) and implantable cardioverter defibrillator (n = 13, 3.1%) and MIDLINE (n = 11, 2.6%). Regarding the topography of UEVT, 90 thromboses (18.2%) affected both deep and superficial veins, 112 thromboses (22.6%) the superficial veins of the arm and 73 (14.8%) thrombosis exclusively forearm veins. The most proximal thrombosed segments were the

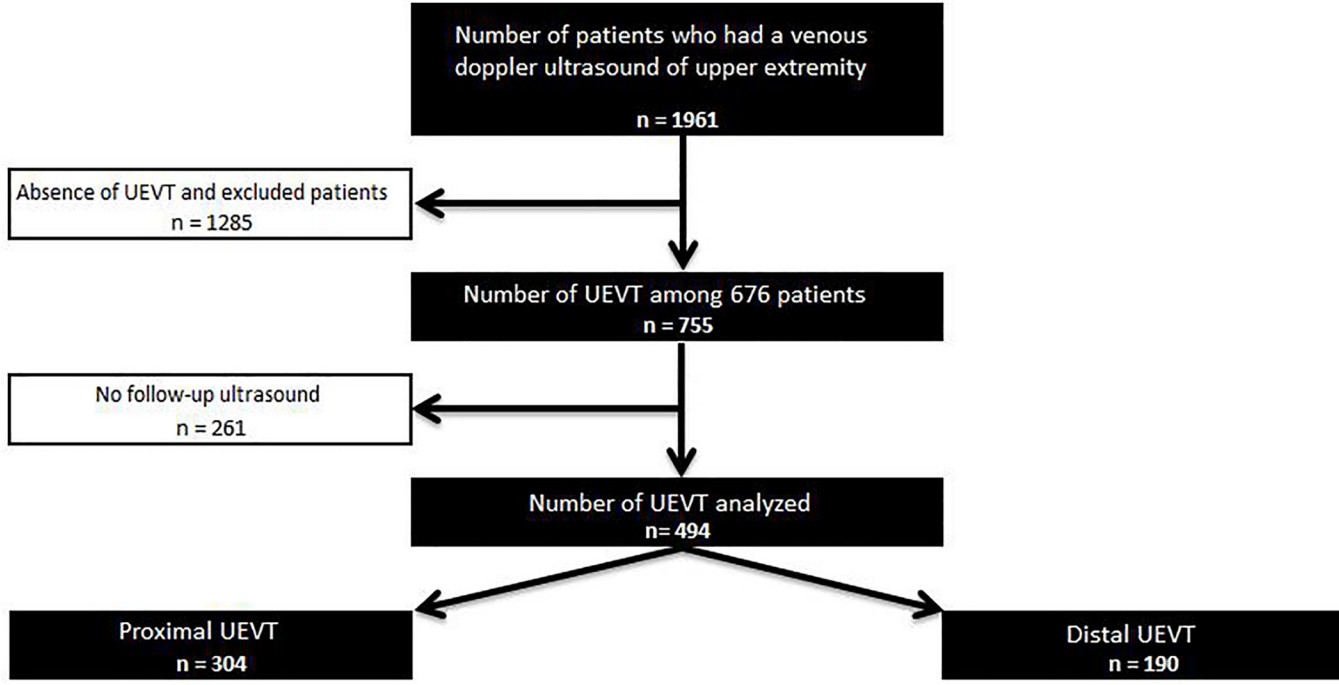

**Fig 1. Flow chart of patient selection included with UE-VT.** (UEVT: Upper Extremity Venous Thrombosis).

innominate veins (n = 106, 21.5%), the internal jugular veins (n = 84, 17.0%), the brachial basilic veins (n = 70, 14, 2%), subclavian veins (n = 67, 13.6%), axillary veins (n = 47, 9.5%), humeral veins (n = 45, 9.1%), brachial cephalic veins (n = 42, 8.5%), antebrachial cephalic veins (n = 26, 5.3%) and the antebrachial basilic veins (n = 7, 1.4%).

Regarding treatments, two patients had angioplasties without stenting, three angioplasties with stenting and a pharmaco-mechanical thrombectomy.

Major and minor Bleedings occurred during follow-up in 44 cases respectively in 25 (5.0%) and in 19 (3.2%) patients; the median times after UEVT were 16 and 30 days respectively.

Two deaths were considered to be related to UEVT (0.4%): none related to PE and 2 related to major bleedings under anticoagulant therapy. A 73-year-old man with gallbladder cancer treated with therapeutic LMWH who have gastrointestinal bleeding and a 49-year-old woman postoperatively after cardiac surgery with ECMO under therapeutic LMWH who have a fatal intracerebral hemorrhage. Neither of these two patients had an over dosage of LMWH.

After UEVT, 469 (94.9%) patients had anticoagulant treatment, 336 (71.8%) had therapeutic LMWH (tinzaparin 175 IU/kg/d, enoxaparin 100 IU/kg twice daily), 45 (9.6%) had prophylactic LMWH (enoxaparin 4,000 IU/d), 34 (6.9%) had NOAC (33 rivaroxaban 20mg/d, 1 apixaban 5mg twice daily) and 72 (14.6%) had VKA. Twenty-five cases remained untreated due to thrombocytopenia <20G/l or bleeding complications. The catheter was removed in 327 (66.2%) cases. The median duration of anticoagulant treatment was 45 days for both patients with hematological malignancies and patients with solid neoplasia. There was no significant difference in the duration of anticoagulant treatment compared to patients without hematological malignancies (p = 0.27) and without solid neoplasia (p = 0.34). The median treatment duration was 77.5 days for deep and proximal UEVT while it was 45.0 days for other UEVT (p<0.0001). For patients with UEVT and a central catheter (PICC line, implantable port, CVC or dialysis catheter), the median treatment duration was 45.0 days for both patients with catheter removal and patients with catheter continuation (p = 0.30).

**Table 1. Characteristics of UEVT depending on their proximal or distal topography.**

| Variables | Total Cohort (UEVT) n = 494 | Proximal Thrombosis (Proximal-VT) n = 304 | Distal Thrombosis (Distal-VT) n = 190 | p |
|---|---|---|---|---|
| **Characteristics** | | | | |
| Mean Age (years) ±SD | 54 ± 18.6 | 54 ± 19.6 | 55 ± 17 | 0.31 |
| Female n (%) | 195 (39.5%) | 123 (40.5%) | 72 (37.9%) | 0.58 |
| Infectious or inflammatory context n (%) | 212 (42.9%) | 129 (42.4%) | 83 (43.7%) | 0.78 |
| Hematological malignancy n (%) | 201 (40.7%) | 110 (36.2%) | 91 (47.9%) | <0.01 |
| Solid cancer n (%) | 70 (14.2%) | 56 (18.4%) | 14 (7.4%) | <0.001 |
| Thoracic Outlet Syndrome n (%) | 12 (2.4%) | 10 (3.3%) | 2 (1.0%) | 0.14 |
| APLS n (%) | 6 (1.2%) | 3 (1.0%) | 3 (1.6%) | 0.68 |
| Pregnancy or PostPartum n (%) | 5 (1.0%) | 2 (0.7%) | 3 (1.6%) | 0.38 |
| **Location of venous thrombosis** | | | | |
| Median number of thrombosed segments [Q1;Q3] | 1[1;2] | 2[1;3] | 1[1;2] | <0.0001 |
| Right laterality n (%) | 327 (66.2%) | 208 (68.4%) | 119 (62.6%) | 0.19 |
| Bilateral n (%) | 9 (1.8%) | 7 (2.3%) | 2 (1.1%) | 0.49 |
| Deep VT n (%) | 349 (70.6%) | 304 (100%) | 45 (23.7%) | <0.0001 |
| **Characteristics of the UEVT** | | | | |
| Asymptomatic n (%) | 60 (12.2%) | 50 (16.4%) | 10 (5.3%) | 0.0002 |
| Endovenous device n (%) | 427 (86.4%) | 263 (86.5%) | 164 (86.3%) | 0.95 |
| Occlusive thrombosis n (%) | 292 (63.9%) | 158 (52.0%) | 134 (70.5%) | <0.0001 |
| PE at diagnosis between D-14 and D-0. n (%) | 11 (2.2%) | 10 (3.3%) | 1 (0.5%) | 0.06 |
| **Treatment of UEVT** | | | | |
| Compression n (%) | 25 (5.0%) | 21 (6.9%) | 4 (2.1%) | 0.02 |
| Symptomatic local care n (%) | 44 (8.9%) | 5 (1.6%) | 39 (20.5%) | <0.0001 |
| VKA n (%) | 72 (15.4%) | 59 (19.4%) | 13 (7.5%) | 0.0003 |
| NOAC n (%) | 34 (7.6%) | 26 (8.6%) | 8 (4.6%) | 0.09 |
| Therapeutic LMWH n (%) | 336 (71.8%) | 213 (72.2%) | 123 (71.1%) | 0.80 |
| UFH n (%) | 49 (10.5%) | 38 (12.5%) | 11 (6.4%) | <0.0001 |
| Prophylactic LMWH n (%) | 45 (9.6%) | 8 (2.6%) | 37 (21.4%) | <0.0001 |
| **Follow Up** | | | | |
| Median follow-up time (days) [Q1-Q3] | 46 [29–90] | 59 [31–104] | 42 [22–57] | <0.0001 |
| Recanalization without venous sequelae n (%) | 188 (38.0%) | 109 (35.9%) | 79 (41.6%) | 0.001 |
| Major Bleeding n (%) | 25 (5.0%) | 14 (4.6%) | 11 (5.6%) | 0.45 |
| Minor Bleeding n (%) | 19 (3.2%) | 14 (4.6%) | 5 (2.7%) | 0.53 |
| PE within 3 months n (%) | 10 (2.0%) | 8 (2.6%) | 2 (1.1%) | 0.37 |
| Death within 3 months n (%) | 39 (7.9%) | 23 (7.6%) | 16 (8.4%) | 0.10 |

(APLS: Antiphospholipid syndrome. NOAC: non-vitamin k antagonist oral anti-coagulants. LMWH: Low Molecular-Weight Heparin. PE: Pulmonary Embolism. SD: Standard Derivation. UE: Upper Extremity. UFH: UnFractionned Heparin. VKA: Vitamin K Antagonists. VT: Vein Thrombosis).

The median time to UEVT recanalization was 100 days; venous occlusion was persistent in 96 (19.4%) at the end of follow-up and 210 (42.5%) had venous recanalization with sequelae. Median recanalization was 43 days [21–85 days] for the 188 (38%) UEVT without sequelae.

Survival with complete UEVT recanalization was significantly different between proximal and distal UEVT (log rank p = 0.001) (Fig 2).

The multivariate Cox model analysis of factors associated with UEVT recanalization without sequelae is presented in Table 2. Anticoagulant treatment durations were not included in

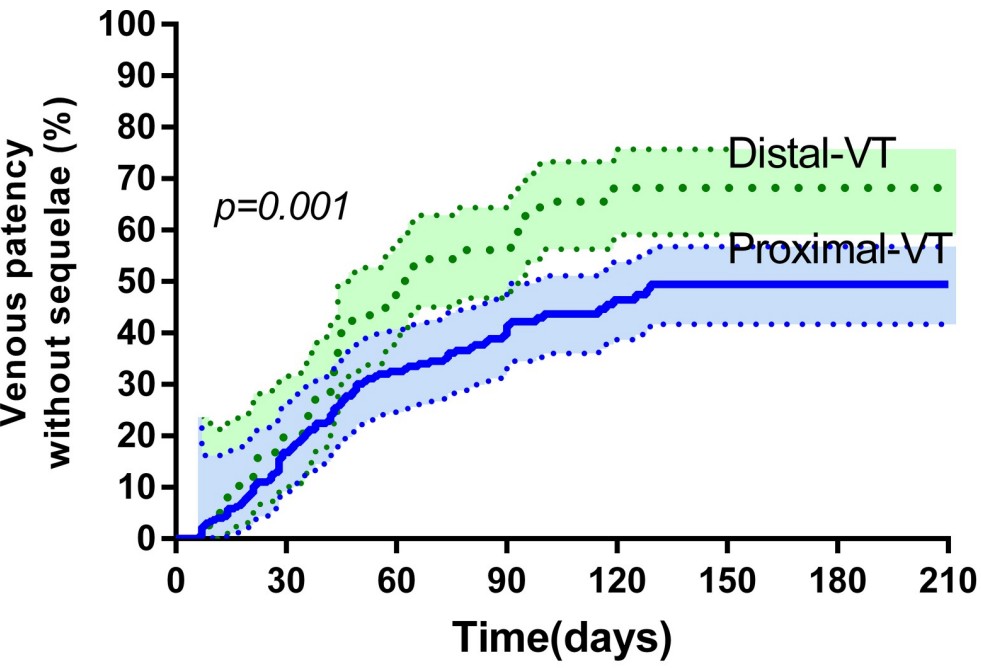

| Nb at risk | 0 | 30 | 60 | 90 | 120 | 150 | 180 | 210 |
|---|---|---|---|---|---|---|---|---|
| Distal-VT | 190 | 120 | 39 | 20 | 11 | 9 | 8 | 8 |
| proximal-VT | 304 | 214 | 139 | 92 | 58 | 41 | 29 | 20 |

**Fig 2. Survival curve of venous patency without sequelae according to UEVT topography.** The percentages of repermeabilization are expressed according to the number of patients still followed at the time of repermeabilization (VT: Venous Thrombosis).

the multivariate analysis since treatment was most often continued if the veins were not recanalized. Fig 3 presents survival curves of venous recanalization without sequelae according to duration of anticoagulant treatment. Patients who received anticoagulant treatment for less than 90 days had significantly more venous recanalization without sequelae than patients treated for more than 90 days (p<0.0001).

The multivariate Cox model analysis of factors associated with deep and proximal UEVT recanalization without sequelae is presented in Table 3. In this group of proximal vein thromboses, non-occlusive UEVT, thrombosis associated with CVC and infectious or inflammatory context were associated with UEVT recanalization without sequelae.

Only 92 UEVT patients were evaluated for post-thrombotic syndrome of upper limbs. Thirteen of these patients (14.1%) had upper extremity post-thrombotic syndrome (UE-PTS) with modified Villalta score ≥4. In 12 (92.3%) cases of UE-PTS, there was a history of deep vein thrombosis.

## Discussion

To our knowledge, this is the first study investigating venous recanalization of UEVT in a large cohort of 494 UEVT. This study showed that clinical context, the type and topography of the thrombosis and the presence of central catheter were associated with recanalization after UEVT. The rate of recanalization without sequelae of UEVT was low (38%) with a median time to recanalization of 43 days.

**Table 2. Cox analysis of factors associated with patency of UEVT without sequelae.**

| Variable | Univariate analysis | | | | Multivariate analysis (recanalization factors without sequelae) n = 457 | |
|---|---|---|---|---|---|---|
| | Without Sequelae n = 188 | With Sequelae n = 306 | HR [95IC] | p | HR [95IC] | p |
| **Characteristics** | | | | | | |
| Mean age (years) ±SD | 54 ± 18.6 | 55 ± 18.7 | 1 [0.99; 1.01] | 0.82 | - | - |
| Female n (%) | 81 (43.1%) | 114 (37.2%) | 1.3 [0.99; 1.77] | 0.06 | † | |
| Infectious or inflammatory context n (%) | 91 (48.4%) | 121 (39.5%) | 1.56 [1.17; 2.08] | <0.01 | | |
| Solid Cancer n (%) | 18 (9.6%) | 52 (17.0%) | 0.64 [0.39; 1.04] | 0.07 | - | - |
| Hematological malignancy n (%) | 94 (50.0%) | 107 (35.0%) | 1.29 [0.97; 1.71] | 0.08 | - | - |
| AVF n (%) | 1 (0.5%) | 11 (3.6%) | 0.13 [0.02; 0.90] | 0.04 | - | - |
| **VT location** | | | | | | |
| Distal UEVT n (%) | 79 (42.0%) | 111 (36.3%) | 1.62 [1.21; 2.18] | 0.001 | - | - |
| Superficial VT of forearm n (%) | 28 (14.9%) | 45 (14.7%) | 1.34 [0.89; 2.01] | 0.16 | 2.79 [1.52; 5.12] | <0.001 |
| Superficial VT of arm n (%) | 78 (41.5%) | 122 (39.9%) | 0.96 [0.72;1.28] | 0.77 | 0.89 [0.57; 1.38] | 0.60 |
| Deep and distal VT n (%) | 51 (27.1%) | 51 (16.7%) | 1.31 [0.95; 1.81] | 0.10 | 1.70 [1.10; 2.63] | 0.02 |
| Deep and proximal VT n (%) | 109 (58.0%) | 195 (63.7%) | 0.62 [0.46; 0.83] | 0.001 | 0.80 [0.50; 1.31] | 0.38 |
| **VT characteristics** | | | | | | |
| Occlusive VT n (%) | 87 (50.9%) | 205 (71.7%) | 0.41 [0.27; 0.62] | <0.0001 | 0.48 [0.34; 0.67] | <0.0001 |
| Short thrombosis (one segment) n (%) | 108 (57.4%) | 159 (52.0%) | 1.8 [1.34; 2.40] | <0.0001 | 1.94 [1.26; 3.00] | 0.003 |
| Endovenous devices n (%) | 165 (87.8%) | 255 (83.3%) | 1.61 [1.01; 2.57] | 0.05 | - | - |
| PICC LINE n (%) | 87 (46.3%) | 90 (29.4%) | 1.55 [1.17; 2.07] | 0.03 | 2.29 [1.48; 3.52] | <0.001 |
| CVC n (%) | 29 (15.4%) | 35 (11.4%) | 1.5 [1.01; 2.23] | 0.05 | 2.40 [1.45; 3.95] | <0.001 |
| **Treatment** | | | | | | |
| Venous compression n (%) | 8 (4.3%) | 17 (5.6%) | 0.46 [0.23; 0.95] | 0.04 | - | - |
| Anticoagulant treatment | 183 (97.3%) | 286 (93.5%) | 0.68 [0.28; 1.67] | 0.40 | - | - |
| Prophylactic LMWH | 19 (10.1%) | 26 (8.5%) | 1.31 [0.81; 2.10] | 0.27 | - | - |
| VKA n (%) | 22 (12.1%) | 50 (17.5%) | 0.59 [0.38; 0.92] | 0.02 | - | - |
| Treatment median (days) [Q1;Q3] | 45 [45; 90] | 60 [45; 90] | 0.98 [0.98; 0.99] | <0.0001 | - | - |

(AVF: Arterio-Venous Fistula, CI: confidence interval, CVC: Central Venous Catheter, HR: Hazard Ratio, LMWH: low molecular weight heparin, PICCLINE: Peripherally Inserted Central Catheter–Line, SD: Standard Derivation, UE: Upper Extremity, VKA: Vitamin K Antagonist, VT: Venous Thrombosis).

†The female gender is a predictive factor for venous patency without sequelae if the context was infectious or inflammatory (HR: 2.02 [1.31; 3.12], p = 0.04).

Forearm superficial-VT (cephalic and basilic veins); deep and distal UEVT, VT affecting only one venous segment and non-occlusive UEVT were factors associated with total recanalization in this study. These better rates of recanalization without sequelae could be explained by the lower thrombus volume in these situations, allowing faster and more efficient thrombus lysis. In contrast, Proximal-VT occurring in larger veins with a larger thrombus would have lower rates of recanalization without sequelae.

Female gender associated with the occurrence of thrombosis in an acute context of infection or inflammation was also a factor associated with better recanalization without sequelae. In contrast, the occurrence of UEVT in a context of solid neoplasia was associated with a lower rate of recanalization. A provoked, transient, modifiable context would logically appear as a factor associated with better recanalization: the disappearance of the transient factor would allow a reduction in pro-thrombotic factors allowing better recanalization. The transient nature is also important for catheter-related thrombosis: PICCLINE and CVC were associated with recanalization rates without sequelae higher than in situations where endovenous device

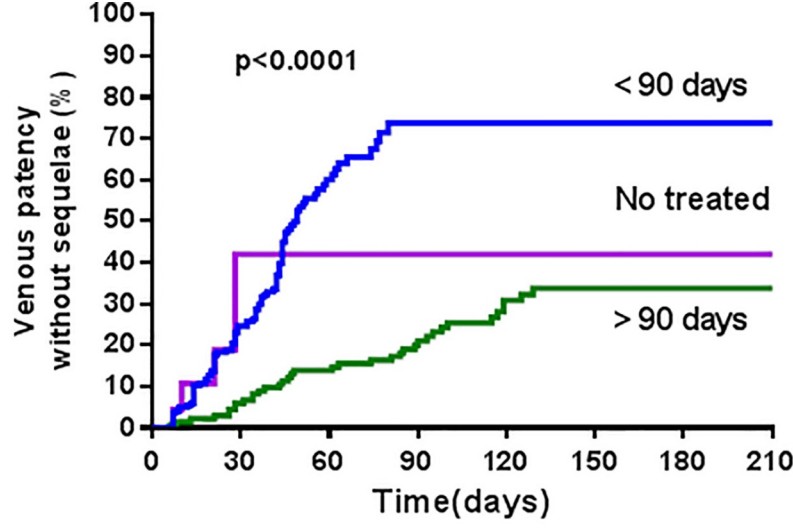

| Nb at risk | 0 | 30 | 60 | 90 | 120 | 150 | 180 | 210 |
|---|---|---|---|---|---|---|---|---|
| no treatment | 25 | 5 | 1 | 1 | 1 | 1 | 1 | 1 |
| < 90 days | 224 | 136 | 31 | 5 | 4 | 4 | 3 | 3 |
| >90 days | 139 | 125 | 103 | 76 | 51 | 35 | 24 | 17 |

**Fig 3. Survival curves of venous recanalization without sequelae according to duration of anticoagulant treatment.**

**Table 3. Cox analysis of factors associated with vein patency of deep and proximal UEVT.**

| Variable | Univariate analysis | | | | Multivariate analysis (recanalization factors without sequelae) n = 289 | |
|---|---|---|---|---|---|---|
| | Without Sequelae n = 109 | With Sequelae n = 195 | HR [95IC] | p | HR [95IC] | p |
| **Characteristics** | | | | | | |
| Mean age (years) ±SD | 51.7±20.0 | 54.8±19.4 | 1.0 [0.99; 1.01] | 0.55 | - | - |
| Female n (%) | 49 (45.0%) | 74 (38.0%) | 0.73 [0.50; 1.07] | 0.11 | - | - |
| Infectious or inflammatory context n (%) | 55 (50.5%) | 74 (38.0%) | 1.90 [1.30; 2.77] | <0.001 | 1.63 [1.10; 2.41] | 0.01 |
| Solid Cancer n (%) | 14 (12.8%) | 42 (21.5%) | 0.69 [0.39; 1.21] | 0.20 | - | - |
| Hematological malignancy n (%) | 48 (44.0%) | 62 (31.8%) | 1.11 [0.76; 1.62] | 0.60 | - | - |
| AVF n (%) | 1 (0.9%) | 4 (2.1%) | 0.34 [0.05; 2.45] | 0.29 | - | - |
| **VT characteristics** | | | | | | |
| Non-occlusive VT n (%) | 42 (40.4%) | 116 (62.7%) | 0.40 [0.27; 0.59] | <0.0001 | 2.23 [1.49; 3.33] | <0.0001 |
| Short thrombosis (one segment) n (%) | 54 (49.5%) | 76 (39.0%) | 1.94 [1.33; 2.89] | <0.001 | - | - |
| Endovenous devices n (%) | 98 (89.9%) | 162 (83.1%) | 1.86 [0.99; 3.47] | 0.05 | - | - |
| PICC LINE n (%) | 42 (38.5%) | 51 (26.1%) | 1.33 [0.9; 1.96] | 0.15 | - | - |
| CVC n (%) | 29 (26.6%) | 35 (18.0%) | 1.94 [1.26; 2.97] | 0.002 | 1.58[1.01; 2.47] | 0.04 |

AVF: Arterio-Venous Fistula, CI: confidence interval, CVC: Central Venous Catheter, HR: Hazard Ratio, LMWH: low molecular weight heparin, PICCLINE: Peripherally Inserted Central Catheter–Line, SD: Standard Derivation, VT: Venous Thrombosis.

was present for a prolonged period. The larger the volume occupied by the catheter in vein lumen, the greater the risk of thrombosis [20], but, in UEVT with endovenous devices, first volume of thrombus is lower and secondly the withdrawal of the catheter would allow the restoration of a greater blood flow and therefore the return to rheological parameters conducive to recanalization. In the case of pace-makers, their removals are rare even in the event of thrombosis, which does not allow the removal of the pro-thrombotic factors necessary for optimal recanalization.

The occurrence of bleeding under anticoagulant therapy, PE or death did not modify venous recanalization in this study. The occurrence of these events is mainly related to the patient's comorbidities [5] but do not directly impact on recanalization.

Topographically, there are similarities between recanalization of LE-VT and UEVT with better recanalization rates in LE-VT affecting only distal and deep segments are described [21]. The female sex also appears to be a factor of better recanalization [22], as well as the absence of comorbidity [23]. The use of NOAC has been shown to be a better treatment than VKA for venous recanalization after deep LE-VT [24]. However, this study does not allow us to compare this result. D-dimer analysis was not performed in our study, yet this biological endpoint associated with the persistence of ultrasound venous sequelae appeared to be a reliable marker for predicting the occurrence of VTED recurrence after deep LE-VT [25].

The limits of this work come from its retrospective design, resulting in a lack of exhaustiveness in the DUS follow-up. Moreover, in this study several factors influenced the duration of anticoagulant treatment, which depended on the practice of each physician, on the presence of active cancer and had little relationship to catheter removal. The topography of venous thrombosis, catheter removal, the presence of active cancer and the presence of venous recanalization on ultrasound controls [26] were the main factors taken into account. Thus, it was difficult to evaluate the role of anticoagulation duration on venous recanalization in this study because this duration was influenced by venous recanalization. The use of VKA, venous compression or long-term anticoagulation treatment were factors associated with the absence of recanalization without sequelae but there is a bias because in this study anticoagulation was most often maintained in the event of persistent sequelae.

Patients with UEVT frequently have thromboses on catheters necessary for their oncologic management. However, in these patients, bleeding risks are high, it is necessary to carry out prospective studies on optimal duration of anticoagulation. Conversely, when the thrombosis occurs with a transient risk factor, shorter anticoagulant treatment could be considered once the factor has resolved, a fortiori, if the thrombosis involves only a venous segment, and it is non-occlusive; again prospective studies are needed.

## Conclusion

In a study on a large cohort of UEVT, the rate of recanalization without sequelae of UEVT was low (38%) with a median time to recanalization of 43 days. The factors significantly associated with recanalization without sequelae were non-occlusive vein thrombosis, UEVT affecting only one segment venous, UEVT associated with PICCLINE or CVC, a superficial thrombosis of the forearm and a deep distal UEVT and female sex in infectious or inflammatory transient context. For deep and proximal UEVT, non-occlusive UEVT, thrombosis associated with CVC and infectious or inflammatory context were factors associated with complete vein recanalization.

Prospective studies in UEVT are needed to assess VTED recurrence, bleeding, venous recanalization and post-thrombotic syndromes.

## Supporting information

**S1 File.**
(XLSX)

## Author Contributions

**Conceptualization:** Gaëtan Ploton, Béatrice Guyomarch, Claire Toquet, Olivier Espitia.

**Data curation:** Gaëtan Ploton, Nicolas Brebion, Marc-Antoine Pistorius, Jérôme Connault, Jeanne Hersant, Alizée Raimbeau, Guillaume Bergère, Mathieu Artifoni, Cécile Durant, Giovanni Gautier, Romain Dumont, Jean-Manuel Kubina, Claire Toquet, Olivier Espitia.

**Formal analysis:** Olivier Espitia.

**Methodology:** Béatrice Guyomarch, Olivier Espitia.

**Project administration:** Olivier Espitia.

**Supervision:** Olivier Espitia.

**Validation:** Olivier Espitia.

**Visualization:** Olivier Espitia.

**Writing – original draft:** Gaëtan Ploton, Béatrice Guyomarch, Claire Toquet, Olivier Espitia.

**Writing – review & editing:** Gaëtan Ploton, Béatrice Guyomarch, Claire Toquet, Olivier Espitia.

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
