## [Decision Letter · Decision Letter 0]

17 Mar 2021

PONE-D-21-05068

Predictive factors of venous recanalization in upper-extremity vein thrombosis.

PLOS ONE

Dear Dr. Espitia,

Thank you for submitting your manuscript to PLOS ONE. After careful consideration, we feel that it has merit but does not fully meet PLOS ONE’s publication criteria as it currently stands. Therefore, we invite you to submit a revised version of the manuscript that addresses the points raised during the review process.

The manuscript is interesting but some concerns raised by the reviewers need to be fixed.

We look forward to receiving your revised manuscript.

Kind regards,

Prof. Raffaele Serra, M.D., Ph.D

Academic Editor

PLOS ONE

Journal Requirements:

'OE received mobility grant, from Société Française de Médecine Vasculaire.'

'The author(s) received no specific funding for this work.'

5. We noticed you have some minor occurrence of overlapping text with the following previous publication(s), which needs to be addressed:

- https://journals.lww.com/md-journal/Fulltext/2020/02070/A_STROBE_cohort_study_of_755_deep_and_superficial.27.aspx

The text that needs to be addressed involves the Introduction.

In your revision ensure you cite all your sources (including your own works), and quote or rephrase any duplicated text outside the methods section. Further consideration is dependent on these concerns being addressed.

Additional Editor Comments:

The manuscript is potentially interesting but some revisions are needed.

Reviewers' comments:

Reviewer's Responses to Questions

**Comments to the Author**

1. Is the manuscript technically sound, and do the data support the conclusions?

Reviewer #1: Yes

2. Has the statistical analysis been performed appropriately and rigorously? 

Reviewer #1: Yes

3. Have the authors made all data underlying the findings in their manuscript fully available?

Reviewer #1: Yes

4. Is the manuscript presented in an intelligible fashion and written in standard English?

Reviewer #1: No

5. Review Comments to the Author

Reviewer #1: I have a number of questions regarding this manuscript:

1. In the abstract, in line 47, occlusive thrombus is associated with complete recanalization (HR 0.48). Then in the conclusion, it states that the “absence of occlusive thrombus” as one of the factors associated with UEVT recanalization. I understand since the HR is 0.48 that the negative of occlusive thrombus is the factor associated with complete recanalization, but I must say it is initially confusing to have it say “occlusive thrombus” in the Results and then the “absence of occlusive thrombosis” in the Conclusion. Can the authors please be consistent in their wording?

2. In the Abstract in line 44, what is the meaning of the term “presence of endovascular material in 86.4%”? Is this scar tissue within the vein from a previous DVT? Is so, then the authors should state this rather than the non-descript terminology they use.

3. On page 4, line 68, when is kidney failure non-serious? I would remove the word serious in front of kidney failure, or change to….kidney failure requiring dialysis.

4. On page 5, line 87, what does the term …” adherent thrombus whose major axis was >5 mm” mean?

5. Can the authors be more complete in the description of the two patients who died “related to major bleedings under anticoagulant therapy” on page 8, lines 158-159? What anticoagulant were they on, where was the bleeding, etc.? This is an important piece of information that they should share with the readers.

6. I was surprised on page 9, line 171-172 that the median treatment time was the same for patients with UEVT and a central catheter who had both catheter continuation and catheter removal. I would expect that with catheter continuation, anticoagulation would have remained intact and not stopped, while with catheter removal, anticoagulation could more readily stop. Can the authors please explain this?

7. I would have expected that an important finding from this study is that the rate of recanalization without squeal of UEVT was low (38%) with a median time to recanalization of 43 days, as reported in the first sentence of the Conclusion. Why is this not included in the abstract? Why is this fact buried in the middle of the Results (page 9, line 175)?

8. Finally, I am concerned about mixing superficial thrombosis in with DVT in this study? Not only are they different entities with different treatment modalities, they are called by different names in the paper…..” superficial thrombosis of the forearm” in the Abstract and “superficial forearm UEVT” in the Conclusion. This is confusing on multiple levels. Would the authors consider moving superficial thrombosis out of the paper, or separating this away from DVT?

6. PLOS authors have the option to publish the peer review history of their article (what does this mean?). If published, this will include your full peer review and any attached files.

Reviewer #1: No

---

## [Author Response · Author response to Decision Letter 0]

8 Apr 2021

Dear Editor,

Please find enclosed a revised manuscript entitled “Predictive factors of venous recanalization in upper-extremity vein thrombosis” for submission to your journal.

All the remarks were taken into account. The answers are detailed in the response to the reviewer file.

Sincerely

Dr Olivier Espitia

Manuscript has been changed, now it respects PLOS ONE style.

'OE received mobility grant, from Société Française de Médecine Vasculaire.'

'The author(s) received no specific funding for this work.'

The changes have been made in the manuscript. Funding information is now in Funding Statement section. The details of these changes were given in the cover letter.

There is no restriction; data were available. All relevant data are within the manuscript and its Supporting Information files.

Manuscript has been changed; ethics statement should only appear in the Methods section.

5. We noticed you have some minor occurrence of overlapping text with the following previous publication(s), which needs to be addressed:

- https://journals.lww.com/md-journal/Fulltext/2020/02070/A_STROBE_cohort_study_of_755_deep_and_superficial.27.aspx

The text that needs to be addressed involves the Introduction.

In your revision ensure you cite all your sources (including your own works), and quote or rephrase any duplicated text outside the methods section. Further consideration is dependent on these concerns being addressed.

Introduction has been changed.

Additional Editor Comments:

The manuscript is potentially interesting but some revisions are needed.

Reviewers' comments:

Reviewer's Responses to Questions

Comments to the Author

1. Is the manuscript technically sound, and do the data support the conclusions?

Reviewer #1: Yes

2. Has the statistical analysis been performed appropriately and rigorously?

Reviewer #1: Yes

3. Have the authors made all data underlying the findings in their manuscript fully available?

Reviewer #1: Yes

4. Is the manuscript presented in an intelligible fashion and written in standard English?

Reviewer #1: No

Manuscript has been changed and native English speaker reviewed the manuscript.

5. Review Comments to the Author

Reviewer #1: I have a number of questions regarding this manuscript:

1. In the abstract, in line 47, occlusive thrombus is associated with complete recanalization (HR 0.48). Then in the conclusion, it states that the “absence of occlusive thrombus” as one of the factors associated with UEVT recanalization. I understand since the HR is 0.48 that the negative of occlusive thrombus is the factor associated with complete recanalization, but I must say it is initially confusing to have it say “occlusive thrombus” in the Results and then the “absence of occlusive thrombosis” in the Conclusion. Can the authors please be consistent in their wording?

Abstract has been changed: Occlusive thrombosis was associated with the absence of UEVT recanalization.

2. In the Abstract in line 44, what is the meaning of the term “presence of endovascular material in 86.4%”? Is this scar tissue within the vein from a previous DVT? Is so, then the authors should state this rather than the non-descript terminology they use.

Endovascular material referred to venous catheter or pacemaker leads. Changes have been made in the manuscript.

3. On page 4, line 68, when is kidney failure non-serious? I would remove the word serious in front of kidney failure, or change to….kidney failure requiring dialysis.

Serious has been deleted and replaced by acute kidney failure or kidney failure requiring dialysis.

4. On page 5, line 87, what does the term …” adherent thrombus whose major axis was >5 mm” mean?

We have clarified the definition of venous blood clot and modified the sentence: UEVT linked to the CVC was defined by a blood clot facing the catheter with blood clot in contact with the vein wall with a length > 5 mm

5. Can the authors be more complete in the description of the two patients who died “related to major bleedings under anticoagulant therapy” on page 8, lines 158-159? What anticoagulant were they on, where was the bleeding, etc.? This is an important piece of information that they should share with the readers.

Indeed, it’s interesting data. These two patients are now described in the manuscript. They had therapeutic LMWH, none had over dosage.

“Two deaths were considered to be related to UEVT (0.4%): none related to PE and 2 related to major bleedings under anticoagulant therapy. A 73-year-old man with gallbladder cancer treated with therapeutic LMWH who have gastrointestinal bleeding and a 49-year-old woman postoperatively after cardiac surgery with ECMO under therapeutic LMWH who have a fatal intracerebral hemorrhage. None had LMWH over dosage.”

6. I was surprised on page 9, line 171-172 that the median treatment time was the same for patients with UEVT and a central catheter who had both catheter continuation and catheter removal. I would expect that with catheter continuation, anticoagulation would have remained intact and not stopped, while with catheter removal, anticoagulation could more readily stop. Can the authors please explain this?

I agree with you about the duration of anticoagulation which was the same depending on whether the catheter was removed or not.

However, there are several factors that explain this lack of difference. On the one hand, the duration of anticoagulation treatment was not standardized and depended on the practice of each physician taking care of these patients. On the other hand, many of these patients had active neoplasia at the time of thrombosis, and the anticoagulant treatments were therefore prolonged because of active treatment of the cancer, whether the catheter was removed or not.

Changes have been made in the manuscript in limits section page 13 line 247.

7. I would have expected that an important finding from this study is that the rate of recanalization without squeal of UEVT was low (38%) with a median time to recanalization of 43 days, as reported in the first sentence of the Conclusion. Why is this not included in the abstract? Why is this fact buried in the middle of the Results (page 9, line 175)?

I thank the reviewer for this remark, I fully agree. We have put forward this data which is now quoted in the abstract and in the first part of the conclusion.

8. Finally, I am concerned about mixing superficial thrombosis in with DVT in this study? Not only are they different entities with different treatment modalities, they are called by different names in the paper…..” superficial thrombosis of the forearm” in the Abstract and “superficial forearm UEVT” in the Conclusion. This is confusing on multiple levels. Would the authors consider moving superficial thrombosis out of the paper, or separating this away from DVT?

We understand the comment made by the reviewer, but this study analyzed all the venous thromboses of the upper limbs, which have never been reported before. It analyzed numerous venous territories: deep and proximal, deep and distal, superficial of the arm and forearm.

However, we agree with the reviewer's opinion on proximal venous thrombosis because of a daily clinical problem. Thus, we performed in 304 patients’ new analyses on the factors associated with venous recanalization studying only deep and proximal venous thrombosis with cox model.

We performed a univariate analysis presented in Table 3 and then a multivariate analysis with a stepwise selection. We started with 6 variables: presence of a central venous catheter, infectious context, number of thrombosed venous segments, context of neoplasia, non-occlusive thrombosis, and sex. The final analysis is presented in Table 3. The presence of a central venous catheter, infectious context, and non-occlusive thrombosis were factors associated with venous recanalization.

I our opinion, data on distal thromboses are also interesting; as there is very little data on superficial vein thrombosis of the upper limbs, we wish to present these data in the manuscript, to better understand the factors associated with their recanalization. Thus, in time, their treatments could be better codified.

It is important to differentiate distal veins of the arm, particularly the basilic veins and the humeral veins, which are frequently used for PICC Line placement, and superficial veins of the forearm which are very often punctured for blood sampling.

We have clarified these notions in the manuscript. When analyzing the factors associated with recanalization, it was only superficial forearm venous thrombosis and not superficial arm venous thrombosis.

Changes have been made in the manuscript and superficial thrombosis of the forearm is now the only formulation used in the manuscript.

---

## [Decision Letter · Decision Letter 1]

15 Apr 2021

PONE-D-21-05068R1

Predictive factors of venous recanalization in upper-extremity vein thrombosis.

PLOS ONE

Dear Dr. Espitia,

Thank you for submitting your manuscript to PLOS ONE. After careful consideration, we feel that it has merit but does not fully meet PLOS ONE’s publication criteria as it currently stands. Therefore, we invite you to submit a revised version of the manuscript that addresses the points raised during the review process.

The manuscript now sounds good. There are some minor revisions before final acceptance.

We look forward to receiving your revised manuscript.

Kind regards,

Prof. Raffaele Serra, M.D., Ph.D

Academic Editor

PLOS ONE

Journal Requirements:

Additional Editor Comments (if provided):

The manuscript is substantially improved. There are only minor revisions now.

Reviewers' comments:

Reviewer's Responses to Questions

**Comments to the Author**

1. If the authors have adequately addressed your comments raised in a previous round of review and you feel that this manuscript is now acceptable for publication, you may indicate that here to bypass the “Comments to the Author” section, enter your conflict of interest statement in the “Confidential to Editor” section, and submit your "Accept" recommendation.

Reviewer #1: All comments have been addressed

2. Is the manuscript technically sound, and do the data support the conclusions?

Reviewer #1: Yes

3. Has the statistical analysis been performed appropriately and rigorously? 

Reviewer #1: Yes

4. Have the authors made all data underlying the findings in their manuscript fully available?

Reviewer #1: Yes

5. Is the manuscript presented in an intelligible fashion and written in standard English?

Reviewer #1: Yes

6. Review Comments to the Author

Reviewer #1: I have two additional comments for the authors.

1. On page 9, line 167 in the revised final version, please change "None had LMWH over dosage" to "Neither of these two patients had an over dosage of LMWH".

2. On page 13, line 248 in the revised final version, please change the abbreviation "PM" to what it actually means.

7. PLOS authors have the option to publish the peer review history of their article (what does this mean?). If published, this will include your full peer review and any attached files.

Reviewer #1: No

---

## [Author Response · Author response to Decision Letter 1]

15 Apr 2021

Dear Editor,

Please find enclosed a revised manuscript entitled “Predictive factors of venous recanalization in upper-extremity vein thrombosis” for submission to your journal.

All the remarks were taken into account. The answers are detailed in the response to the reviewer file.

Sincerely

Dr Olivier Espitia

Review Comments to the Author

Reviewer #1: I have two additional comments for the authors.

1. On page 9, line 167 in the revised final version, please change "None had LMWH over dosage" to "Neither of these two patients had an over dosage of LMWH".

The changes have been made in the manuscript.

2. On page 13, line 248 in the revised final version, please change the abbreviation "PM" to what it actually means.

PM means pace-maker, the abbreviation has been changed in the manuscript.

---

## [Editor Report · Decision Letter 2]

23 Apr 2021

Predictive factors of venous recanalization in upper-extremity vein thrombosis.

PONE-D-21-05068R2

Dear Dr. Espitia,

We’re pleased to inform you that your manuscript has been judged scientifically suitable for publication and will be formally accepted for publication once it meets all outstanding technical requirements.

Kind regards,

Raffaele Serra, M.D., Ph.D

Academic Editor

PLOS ONE

Additional Editor Comments (optional):

Amended manuscript is acceptable.
---

## [Editor Report · Acceptance letter]

27 Apr 2021

PONE-D-21-05068R2 

Predictive factors of venous recanalization in upper-extremity vein thrombosis. 

Dear Dr. Espitia:

I'm pleased to inform you that your manuscript has been deemed suitable for publication in PLOS ONE. Congratulations! Your manuscript is now with our production department. 

Kind regards, 

on behalf of

Prof. Raffaele Serra 

Academic Editor

PLOS ONE